

# Anatomical notes and discussion of the first described aetosaur *Stagonolepis robertsoni* (Archosauria: Suchia) from the Upper Triassic of Europe, and the use of plesiomorphies in aetosaur biochronology

William G. Parker

Division of Science and Resource Management, Petrified Forest National Park, Petrified Forest, AZ, United States of America

## ABSTRACT

*Stagonolepis robertsoni*, from the Late Triassic of Scotland, was the first named aetosaurian. Known mostly from a series of natural molds from two localities, the osteology of this taxon has been difficult to interpret. Detailed work on this material in the late 1950s resulted in a monograph that set the standard for the understanding of aetosaurians, making *Stagonolepis robertsoni* the best known aetosaurian; however, little has been done with this material since. Reanalysis of this material shows that despite its limitations the early 1960s reconstruction work depicts the preserved characteristics faithfully, especially in the skull. The first cervical rib is extremely anteroposteriorly elongate as in *Alligator*, a character not previously recognized in aetosaurians. Diapophyseal and zygapophyseal vertebral laminae are present in the cervical and trunk vertebrae. The ilium is autapomorphic with distinct pre- and post-processes of the iliac blade. The osteoderms differ from North and South American material that has been ascribed to the genus. Those assignments are based on plesiomorphies within Aetosauria, such as a radial ornamentation and a posteriorly located and medially offset dorsal eminence. Biostratigraphic correlations using taxonomic conclusions based on plesiomorphic characters should not be used. The holotype specimen of *S. robertsoni* is currently diagnostic, in part because ventral osteoderms are not known for many aetosaurian taxa and the surface ornamentation of randomly distributed, closely packed oblong pits found in *S. robertsoni* is unique within Aetosauria.

## INTRODUCTION

Aetosaurs are exclusively Late Triassic quadrupedal, heavily armored pseudosuchians known globally with the exception of some parts of Gondwana: southern Africa, Antarctica, and Australia. The most recent overview of this clade is by *Desojo et al. (2013)*, with older material and new taxa recently described by (*Heckert et al., 2015*; *Heckert, Fraser & Schneider, 2017*; *Parker, 2016b*; *Parker, 2018*; *Schoch & Desojo, 2016*). The first described

Corresponding author
William G. Parker,
William_Parker@nps.gov

aetosaurian fossil was christened *Stagonolepis robertsoni* (*Agassiz, 1844*) for a series of ventral osteoderms. Because the unit they were discovered in was thought to be the Devonian 'Old Red Sandstone', these osteoderms were considered to represent a large ganoid fish (*Agassiz, 1844*). Subsequent discoveries (detailed in *Walker, 1961*) demonstrated that these remains belonged instead to a stem-crocodylian, and that the rocks they were found in were instead more likely Triassic in age (*Huxley, 1859*; *Huxley, 1869*; *Huxley, 1875*; *Huxley, 1877*). The original materials worked on by Louis Agassiz and T.H. Huxley, as well as additional discoveries made in the 1920s and 1930s, were the subject of an extensive monograph (*Walker, 1961*). This work was influential because it finally established that aetosaurians were a globally distributed Late Triassic group distinct from phytosaurs.

Although since that time the only new material described for *Stagonolepis robertsoni* is a braincase (MCZD 2–4; *Gower & Walker, 2002*), *S. robertsoni* is still considered to be the 'model' aetosaurian (*Desojo et al., 2013*) and the taxon is featured prominently in phylogenetic and biomechanical analyses (e.g., *Parrish, 1994*; *Heckert, Hunt & Lucas, 1996*; *Heckert & Lucas, 1999*; *Parker, 2007*; *Desojo & Vizcaíno, 2009*; *Parker, 2016a*). However, since 1961 the taxon has never been adequately re-evaluated despite the great strides that have been made in our understanding of aetosaurian morphology and systematics, particularly the recognition that armor surface ornament, width-length ratios, and lateral osteoderm morphology are diagnostic to clades, and in some cases, to species (e.g., *Long & Ballew, 1985*; *Long & Murry, 1995*; *Heckert & Lucas, 2000*; *Parker, 2007*; *Martz & Small, 2006*). Other named taxa (*Calyptosuchus wellesi* *Long & Ballew, 1985*, *Ebrachosaurus singularis* *Kuhn, 1936*, *Aetosauroides scagliai* *Casamiquela, 1960*; *Argentinosuchus bonapartei* *Casamiquela, 1960*) have been assigned to the genus *Stagonolepis* based on the presence of a radial dorsal paramedian osteoderm ornamentation and a width/length ratio of 2.5/1 (e.g., *Long & Murry, 1995*; *Heckert & Lucas, 2000*; *Heckert & Lucas, 2002*); however, these assignments have been questioned or refuted by other workers (e.g., *Desojo & Ezcurra, 2011*; *Parker, 2018*). A closely related species, *Stagonolepis olenkae* (*Sulej, 2010*), has been described from Poland and its distinctness from *S. robertsoni* is debated (*Lucas, Spielmann & Hunt, 2007*; *Antczak, 2016*); however, it is treated here as distinct as discussed by *Parker (2016a)*.

The purpose of this paper is to reanalyze referred material of *Stagonolepis robertsoni* present at the Natural History Museum, London and evaluate it in context of our current understanding of aetosaurian anatomy. The material is not fully redescribed here as the original descriptions by *Walker (1961)* and *Gower & Walker (2002)* are still adequate; instead I focus on characteristics whose taxonomic significance were not completely recognized by Walker given the understanding of aetosaurians at the time of his monograph. These newly recognized characteristics are useful for examining the phylogenetic relationships of *S. robertsoni* and other aetosaurians and further demonstrates the importance of redescribing historic type specimens (*Parker, 2013*).

## MATERIALS

Of the materials of *S. robertsoni* that were available to *Huxley (1859)*, *Huxley (1875)* and *Huxley (1877)* those available today at the Natural History Museum, London consist only of a couple of now chipped casts of a left femur (NHMUK PV R 581) and a paramedian osteoderm and phalanx (NHMUK PV R 582). The rest of the material is in other institutions such as the Geological Survey Museum, Keyworth and the Elgin Museum in Scotland (*Walker, 1961*; Walker undated and unpublished notes at the NHMUK). More prominent materials at the NHMUK consist almost entirely of large blocks of a highly indurated sandstone that preserve natural molds of various bones. Painted casts of these were created in 1885 and presented to the museum by the Reverend George Gordon. These specimens allow study of portions of the bones in two dimensions, and were available for study to Charles Camp in 1935 (Camp unpublished notes at the UCMP, 1935) and Alick Walker in the 1950s (*Walker, 1961*). *Walker (1961)* also created a series of PVC casts (e.g., NHMUK PV R4787) from the original sandstone molds that preserved more detail and allowed for 3D visualization of the bones. These PVC casts are still in the NHMUK collections although many have deteriorated significantly.

For this study the following blocks/specimens were examined and photographed: MCZD2 (5 parts), partial skull and articulated cervical osteoderms; NHMUK PV R 581, cast of a left femur; NHMUK PV R 582, cast of a paramedian osteoderm and a phalanx; NHMUK PV R 4784a, cast of a slab with a basioccipital and associated articulated series of cervical and trunk vertebrae, including the cervical ribs, scapulocoracoid, partial humerus; NHMUK PV R 4785a, cast of a slab with osteoderms and ribs; NHMUK PV R 4786a, cast of a slab with anterior caudal centra and ribs in articulation, and the distal end of a left tibia; NHMUK PV R 4787; sandstone block containing the natural mold of much of a skull plus an associated PVC cast; NHMUK PV R 4787a,a cast of a specimen showing the lower portion of the left side of a good skull, especially the mandible; NHMUK PV R 4789a, cast of a slab with ribs, osteoderms, a right ilium, and a partial maxilla; NHMUK PV R 4790a, cast of a slab with osteoderms, a right ilium, and an ischium; NHMUK PV R 4797a, a cast with lower limb bones and appendicular osteoderms; NHMUK PV R 4799, a sandstone block with an impression of an anterior cervical vertebra; NHMUK PV R 8586, a cast of ELGNM 38R, the left side of internal surface of snout (figured by *Huxley, 1877*); NHMUK PV R 27404, a cast of the holotype specimen that consists of a segment of the articulated ventral carapace; NHMUK PV R 36392, a sandstone slab with osteoderms, including six articulated left lateral osteoderms; NHMUK PV R 36394; a sandstone slab with the impressions of articulated ventral osteoderms.

An attempt was made to use the CT scanner at the NHMUK to scan the negative space in block NHMUK PV R 4787 to get a three-dimensional view of the skull, but unfortunately that scanner could not penetrate the hard sandstone matrix of the block.

# SYSTEMATIC PALEONTOLOGY

**Archosauria** *Cope, 1869* **sensu** *Gauthier & Padian, 1985*
**Pseudosuchia** *Zittel, 1887–1890* **sensu** *Gauthier & Padian, 1985*
**Aetosauria** *Marsh, 1884* **sensu** *Parker, 2007*
**Desmatosuchia sensu** *Parker, 2016a*
**Stagonolepidinae sensu** *Heckert & Lucas, 2000*
*Stagonolepis Agassiz, 1844*
*Stagonolepis robertsoni Agassiz, 1844*

*1844* Stagonolepis robertsoni: *Agassiz*, p. 139, pl. XXXI, figs. xiii, xiv
*1859* Stagonolepis robertsoni: *Huxley*, p. 440, pl. XIV, figs. 1–3
*1877* Stagonolepis robertsoni: *Huxley*, p. 1, pl. I-X.
*1902* Staganolepis [sic] robertsoni: *Huene*, p. 54, figs. 62-0–67, 72, 73.
*1908* Stagonolepis robertsoni: *Huene*, p. 392, figs. 347–348.
*1936* Stagonolepis robertsoni: *Huene*, p. 207, fig. 3.
*1942* Stagonolepis robertsoni: *Huene*, p. 223, figs. 45–49.
*1961* Stagonolepis robertsoni: *Walker*, p. 103, figs. 2–23, 24b, 25b, pl. 9–12.
*1976* Stagonolepis robertsoni: *Krebs*, p. 40, figs. 3, 4, 9, 10d, 12, 15, 16, 17c-e, 19d-e, 20d-e, 26b, 27.
*1978* Staganolepis [sic]: *Bonaparte*, p. 300, figs. 137b, 138.
*1986* Stagonolepis: *Parrish*, p. 8, figs. 6, 14c3.
*1988* Stagonolepis: *Carroll*, p. 273, figs. 13.15, 13.16.
*1988* Stagonolepis: *Fraser*, p. 132, fig. 5b.
*1991* Stagonolepis: *Sereno*, p. 11, figs. 10, 27f.
*1996* Stagonolepis: *Lucas & Heckert*, p. 57, fig. 4.
*2000* Stagonolepis robertsoni: *Heckert & Lucas*, p. 1552, figs. 4c, e.
*2001* Stagonolepis robertsoni: *Lucas & Heckert*, p. 719, figs. 2, 3.
*2002* Stagonolepis robertsoni: *Gower & Walker*, p. 7, figs. 1–4, 6.
*2010* Stagonolepis robertsoni: *Sulej*, p. 878, figs. 8a, 9f.
*2011* Stagonolepis robertsoni: *Desojo & Ezcurra*, p. 599, figs. 3d-f, 7b.
*2013* Stagonolepis robertsoni: *Desojo et al.*, p. 207, figs. 3e-f, 4g, 5a-h, 6a-f, 7c-f.
*2016a* Stagonolepis robertsoni: *Parker*, p. 32, figs. 1, app. B, fig. 11j.

*Holotype*—ELGNM 27R, impression of a segment of the ventral carapace (*Agassiz, 1844*). NHMUK PV R 27404 is a negative cast of this specimen.

*Referred Material*—see Materials for a list of NHMUK specimens.

*Occurrence*—Lossiemouth Sandstone Formation, Moray, Scotland, U.K (*Walker, 1961*).

*Age*—Late Triassic, late Carnian to early Norian (*Benton & Walker, 2011*).

*Diagnosis*—*S. robertsoni* is diagnosed by the following autapomorphies: ventral osteoderms rectangular with randomly arranged, oblong pits; first cervical vertebra with elongate cervical ribs that extend back to the position of the 4th cervical vertebra; posterior

process of iliac blade forms an acute angled tip; anterior process of the iliac blade is anteroposteriorly short, dorsoventrally thin, and ventrally hooked.

*S. robertsoni* can also be distinguished by the following combination of character states: premaxilla with four, possibly five teeth as in *Stagonolepis olenkae* and *Paratypothorax andressorum Long & Ballew, 1985*, *Aetosaurus ferratus Fraas, 1877*, and *Neoaetosauroides engaeus Bonaparte, 1978* (differs from three in *Stenomyti huangae Small & Martz, 2013*, and the absence of premaxillary teeth in *Desmatosuchus smalli Parker, 2005*); premaxillary tip expanded laterally as in *Stagonolepis olenkae*, *Desmatosuchus smalli*, and *Neoaetosauroides engaeus* (absent in *Aetosaurus ferratus*, *Stenomyti huangae*; *Paratypothorax andressorum*); distinct ridge on lateral side of maxilla beneath the antorbital fossa as in *Paratypothorax andressorum* and *Stenomyti huangae* (this ridge is absent in *Desmatosuchus smalli*, *Longosuchus meadei* (*Sawin, 1947*), *Neoaetosauroides engaeus*, and *Stagonolepis olenkae*); long axis of the jugal anterodorsally inclined as in *Desmatosuchus spurensis* (*Case, 1920*) and *Longosuchus meadei* (the ventral margin of the jugal is level in *Aetosaurus ferratus*, *Paratypothorax andressorum*, and *Stenomyti huangae*); teeth thecodont with a swollen bases and non-recurved tips as in *Desmatosuchus smalli* (differs from *Aetosauroides scagliai* and *Aetosaurus ferratus*); parabasisphenoid elongate, with anteroposteriorly separated basal tubera and basipterygoid processes as in *Neoaetosauroides engaeus*, *Aetosaurus ferratus*, and *Desmatosuchus smalli* (differs from *Scutarx deltatylus Parker, 2016a*, *Paratypothorax andressorum, Tecovasuchus chatterjeei Martz & Small, 2006*, and *Desmatosuchus spurensis*); mandible "slipper-shaped" with ventrolateral portion of the splenial visible in lateral view beneath the ventral margin of the dentary as in all aetosaurs except for *Aetosauroides scagliai* and *Typothorax coccinarum Cope, 1875*; posterior cervical vertebrae with a ventral keel as in *Aetosauroides scagliai*, *Calyptosuchus wellesi*, *Sierritasuchus macalpini Parker, Stocker & Irmis, 2008*, *Neoaetosauroides scagliai*, and *Scutarx deltatylus* (keels absent in *Longosuchus meadei*, *Desmatosuchus spurensis*, and *Aetobarbakinoides brasiliensis Desojo, Ezcurra & Kischlat, 2012*); pubis with two obturator foraminae (convergent in *Scutarx deltatylus*); trunk and anterior caudal paramedian osteoderms with a length/width ratio of about 2.5:1 as in *Desmatosuchus spurensis*, *Longosuchus meadei*, and *Aetobarbakinoides brasiliensis* (differs from the much wider paramedians of *Typothorax coccinarum* and *Paratypothorax andressorum*); raised anterior bar on osteoderms as in all non-desmatosuchin (*sensu Parker, 2016a*) aetosaurs; anteromedial and anterolateral projections of the anterior bar present on the trunk osteoderms as in all non-desmatosuchin aetosaurs; anterolateral projection of the anterior bar not elongate as in *Stagonolepis olenkae*, *Aetosaurus ferratus*, and *Typothorax coccinarum* (differs from *Adamanasuchus eisenhardtae Lucas, Hunt & Spielmann, 2007* and *Scutarx deltatylus*); trunk paramedian osteoderms lack ornament along the posterior edge in the region of the dorsal eminence as in *Aetosauroides scagliai*; dorsal eminence of trunk paramedian osteoderms situated on the posterior osteoderm margin and offset medially (as in most aetosaurs except *Desmatosuchus* and *Paratypothorax andressorum*); dorsal surface ornament of the paramedian and lateral osteoderms anastomosing (interconnected series of radiating ridges surrounding subrounded and subrectangular pits, *Taborda, Heckert & Desojo, 2015*); lateral osteoderms are nearly equant with distinct lateral and dorsal flanges

as in all non-desmatosuchin aetosaurs; lateral osteoderms lack pronounced horns or spines as in most aetosaurs outside of Typothoracisinae and Desmatosuchini.

## DESCRIPTION

### Skull

Block NHMUK PV R 4787a is a cast of the lower portion of a skull in semi-articulation including much of the lower jaw, quadrate, portions of the palate and maxilla, and the premaxilla (Fig. 1A). These elements represent the left side of the skull, so the cast provides an internal (medial) view. The semi-articulated condition allows for the determination of the skull length, which from the retroarticular process to the tip of the premaxilla is about 240 mm. The dentigerous elements show the presence of at least eight dentary, four maxillary, and four premaxillary teeth. Alick Walker's PVC cast of this specimen shows even more details (Fig. 1B) including the upper portions of the skull and the braincase, especially the parabasisphenoid. Thus block NHMUK PV R 4787 is the natural mold of a nearly complete skull of *S. robertsoni* (*Walker, 1961*). Details of this specimen that are not visible in the cast NHMUK PV R 4787a include: (1) a possible 5th premaxillary tooth crown, or alternatively the tip of the right premaxilla in articulation with the left (Fig. 1B); (2) a portion of the right squamosal and impressions of the skull roof, (3) the left quadrate in articulation with the left articular, and (4) impressions of the braincase, especially the basisphenoid including the left and right basitubera and basipterygoid processes, and the cultriform process is also preserved.

A cast of the right maxilla from NHMUK PV R 4787 (Fig. 2) shows that overall the element is more slender than that of *Stagonolepis olenkae* (*Sulej, 2010*) with at least six alveoli (Fig. 2B), but the anterior portion is covered. A very distinct transverse ridge is present on the lateral surface along the anterior and ventral borders of the antorbital fossa, making the fossa extremely pronounced; this ridge also occurs in *Stenomyti huangae* (*Small & Martz, 2013*) and *Paratypothorax andressorum* (*Schoch & Desojo, 2016*), but is not present in *Stagonolepis olenkae* (*Sulej, 2010*), *Desmatosuchus* (*Case, 1922*; *Small, 2002*), *Longosuchus* (*Parrish, 1994*), or *Neoaetosauroides* (*Desojo & Báez, 2007*). The medial side of the maxillary body is marked by an elongate medial shelf that articulates with the palatal bones (*Walker, 1961*). The articulations with the lacrimal and jugal are each complex, with regions of overlap between the two bones; this is also visible in *Longosuchus meadei* (TMM 31185-98), *Stagonolepis olenkae* (ZPAL AbIII/2454/3), and *Aetosauroides scagliai* (USFM 11050), and probably present in other aetosaurs as well but obscured in most articulated skulls. Finally, a pneumatic accessory cavity is present on the medial size of the ascending process as described for *Desmatosuchus smalli* (*Small, 2002*).

NHMUK PV R 8586 is a cast of ELGNM 38R (Fig. 3), which was figured by *Huxley (1877)* and features the left side of the internal portion of the snout, including the premaxilla, maxilla, and nasals. The premaxilla measures 63 mm in length and bears four teeth (Fig. 3) with an edentulous anterior portion typical for aetosaurians; four or more teeth also occur in *Aetosaurus ferratus* (*Schoch, 2007*) *Stagonolepis olenkae* (*Sulej, 2010*), *Neoaetosauroides engaeus* (*Desojo & Báez, 2007*), and *Paratypothorax andressorum* (*Schoch & Desojo, 2016*),

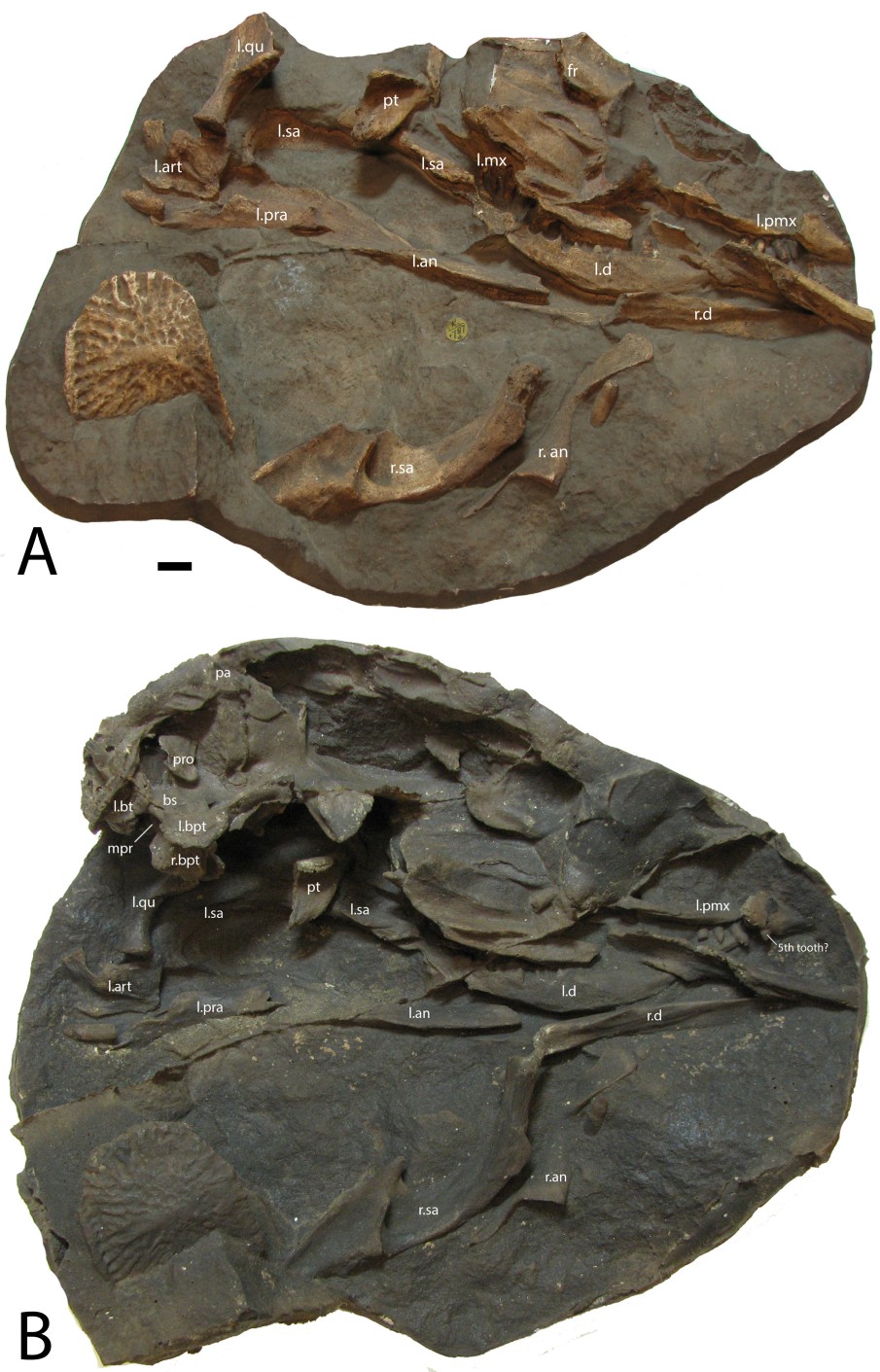

**Figure 1 Casts of bones of *Stagonolepis robertsoni.* (A)** NHMUK PV R 4787a, 1885 cast which represents much of the lower portion of a skull. (B) NHMUK PV R 4787, PVC cast of the same specimen that shows more details of the braincase and dorsal portion of the skull. Scale bars equal 1 cm. Abbreviations: an, angular; art, articular; bpt, basipterygoid process; bs, parabasisphenoid; bt, basal tuber; d, dentary; fr, frontals; l, left; mx, maxilla; mpr, area of the medial pharyngeal recess; pa, parietal; pmx, premaxilla; pra, prearticular; pro, prootic; pt, pterygoid; qu, quadrate; r, right; sa, surangular. Elements from the left side have a prefix of l.; elements of the right side have a prefix of r.

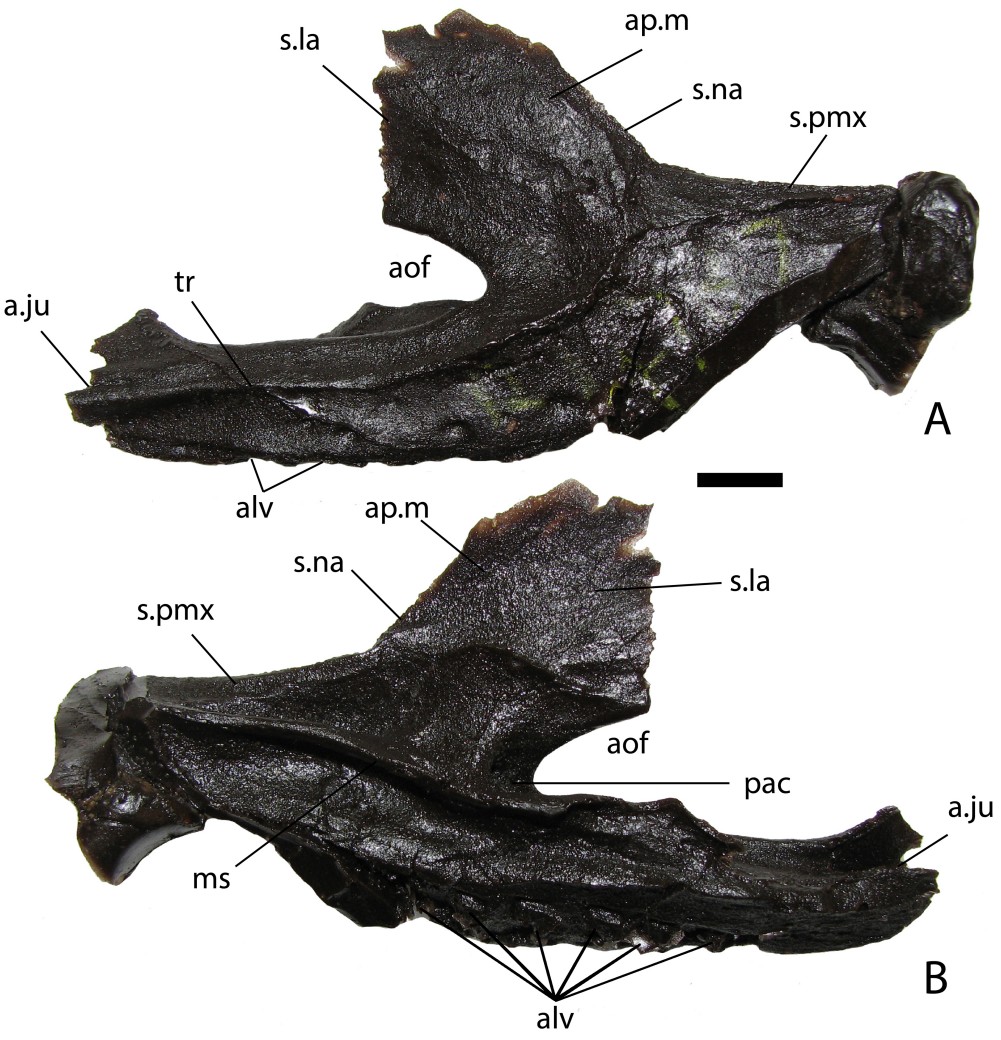

**Figure 2** **NHMUK PV R 4787, cast of a right maxilla of *Stagonolepis robertsoni.*** Lateral (A) and medial (B) views. Scale bar equals 1 cm. Abbreviations: a., articulation with listed element; alv, alveoli; aof, antorbital fenestra; ap.m, ascending process of the maxilla; ju, jugal; la, lacrimal; ms, medial shelf; na, nasal; pac, pneumatic accessory cavity; pmx, premaxilla; tr, transverse ridge.

whereas *Stenomyti huangae* only possesses three (*Small & Martz, 2013*), and *Desmatosuchus* lacks any (*Small, 2002*). The maxilla bears six teeth as preserved, but it is missing the posterior portion. The external naris is 22 mm at its deepest point and 72 mm in length, but missing the posteriormost section. The nasal has a pronounced ridge on the medial edge, which at a position just dorsal to the 3rd premaxillary tooth migrates ventrally to the ventral margin of the nasal where it contacts the premaxilla. The anterior tip of the premaxilla bears a prominent ridge that divides the element into a flat surface that slopes into the external naris, and a second triangular area that slopes anteroventrally (Fig. 3). This ridge is the anterior expansion and also occurs in other aetosaurians such as *Desmatosuchus smalli* (*Small, 2002*), *Neoaetosauroides engaeus* (PVL 4363), and differs from

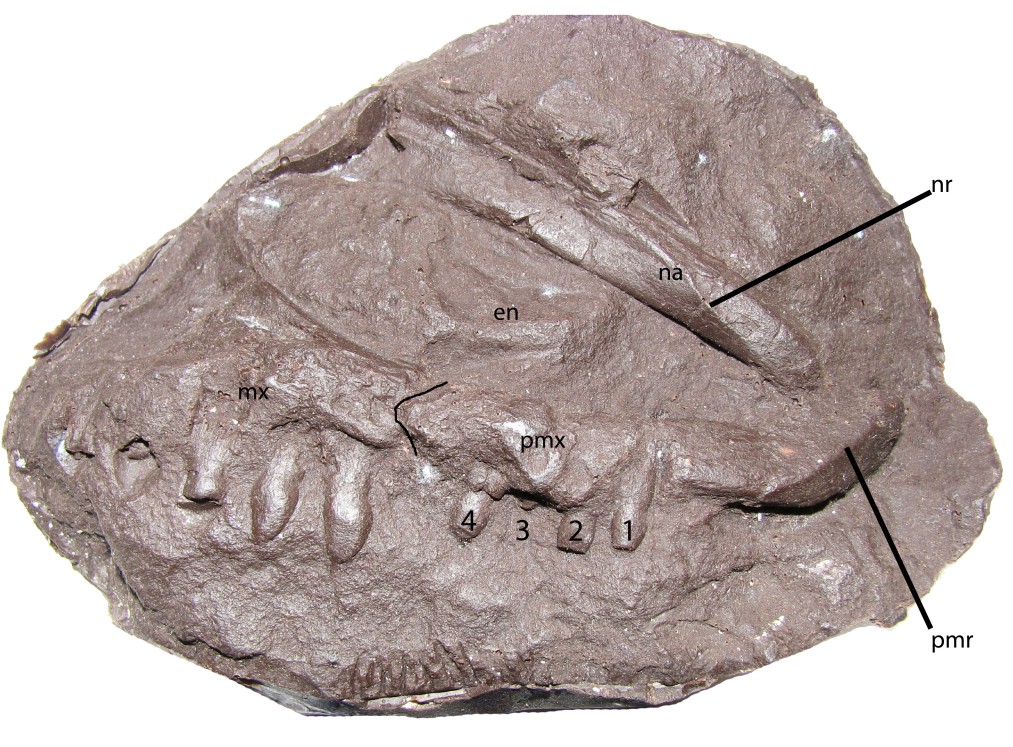

**Figure 3** **NHMUK PV R 8586, skull of *Stagonolepis robertsoni*.** Cast of the anterior part of the right side of the skull of *S. robertsoni* in lateral view. Scale bar equals 1 cm. Abbreviations: en, external naris; mx, maxilla; na, nasal; nr, nasal ridge; pmr, premaxillary ridge; pmx, premaxilla.

aetosaurians such as *Aetosaurus ferratus* (*Schoch, 2007*), *Aetosauroides scagliai* (PVL 2059), and *Typothorax coccinarum* (YPM 58121). The premaxilla bears a small dorsal protuberance above the first tooth position that extends dorsally into the external naris (*Walker, 1961*), which also occurs in *Stagonolepis olenkae* (*Sulej, 2010*) and *Desmatosuchus smalli* (*Small, 2002*); however, in *S. olenkae* it is dorsal to the second tooth position (*Antczak, 2016*) and *D. smalli* has an edentulous premaxilla (*Small, 2002*). There is only a slight swelling in this position in *Stenomyti huangae* (*Small & Martz, 2013*).

The best-preserved skull material is MCZD 2, which consists of seven small blocks that fit together to present much of the skull and the anterior section of the neck (*Walker, 1961*: figs. 26–29; *Gower & Walker, 2002*: fig. 1). The material consists of well-preserved bone and is not a natural mold as is most of the *S. robertsoni* material. This specimen has previously been described in great detail so this will not be duplicated here (*Walker, 1961*; *Gower & Walker, 2002*). However, notable is that the basitubera and basipterygoid processes are widely separated anteroposteriorly from each other so that the parabasisphenoid is elongate, supporting what can be observed in NHMUK PV R 4787. *Neoaetosauroides engaeus* (PVL 5698; *Desojo & Báez, 2007*), *Aetosaurus ferratus* (*Schoch, 2007*), *Paratypothorax andressorum* (*Schoch & Desojo, 2016*), and the basal taxon *Aetosauroides scagliai* (PVSJ 326) also have an

elongate parabasisphenoid, so that is the plesiomorphic state within Aetosauria. This differs significantly from the condition in *S. olenkae*, where the basitubera and basipterygoid processes nearly contact (*Sulej, 2010*:figs. 1D, 1F) and thus the parabasisphenoid is anteroposteriorly short. In *Scutarx deltatylus* (*Parker, 2016b*:fig. 7) and *Desmatosuchus spurensis* (UMMP 7476) the parabasisphenoid is also anteroposteriorly short; whereas the condition in *Calyptosuchus wellesi* is unknown (*Parker, 2018*).

## Presacral vertebrae and ribs

In block NHMUK PV R 4784a, which is a cast created from a natural mold and represents the postcranial skeleton of the skull NHMUK PV R 4787 (*Walker, 1961*), the occipital condyle and left paroccipital process of the skull are present (Fig. 4A). If the cervical count is nine (as in *Desmatosuchus spurensis*), the four anterior cervicals are missing (including the axis and atlas); however, cervical ribs are present for three of these positions. Especially striking are two very elongate posteriorly projecting cervical ribs underlying the ventral surfaces of the two more posteriorly positioned ribs (Fig. 4A). The elongate ribs (left and right sides) originate where the axis/atlas would be located and are very similar to the greatly elongate cervical ribs found in *Alligator* (*Reese, 1915*). As exposed these ribs measure 85 mm in length (the ends are covered), more than three times the lengths of the other exposed cervical ribs. The elongate ribs were not noted by *Walker (1961)* and have not previously been described for any aetosaur. The axis/atlas and third cervical are present in MCZD 2, but unfortunately are very poorly preserved; however, this block also preserves a very elongate first cervical rib.

The first well-preserved cervical vertebra in NHMUK PV R 4784a is the 8th (*Walker, 1961*), which is visible in right lateral view and preserves details of the centrum and much of the neural arch and transverse process (Fig. 4B). A disarticulated cervical rib is present across the centrum and probably does not belong to this vertebra. The centrum measures 25 mm in length and the ventral surface is strongly keeled. Keels occur on the ventral surfaces of the cervical vertebrae in *Aetosauroides* (*Desojo & Ezcurra, 2011*), *Neoaetosauroides* (*Desojo & Báez, 2007*), and *Scutarx* (*Parker, 2016a*; *Parker, 2016b*) while the ventral surfaces of the cervicals lack keels in *Longosuchus* (*Long & Murry, 1995*) and *Desmatosuchus* (*Case, 1922*; *Parker, 2018*), and *Aetobarbakinoides* (*Desojo, Ezcurra & Kischlat, 2012*). The condition may be variable in *Typothorax*; ventral keels are absent in the cervicals of the small specimen described by *Martz (2002)* but present in at least some larger specimens (*Heckert et al., 2010*), suggesting there may be allometric or ontogenetic variability. The parapophysis in the 8th cervical of NHMUK PV R 4784a is low on the anterior rim of the centrum, but not completely at the base. The transverse process projects laterally and slightly ventrally, is 25 mm long, and bears a flaring sub-rectangular head in lateral view. A distinct posterior centrodiapophyseal lamina (*Wilson, 1999*) stretches from the base of the transverse process to the posterior portion of the neurocentral suture. This is the "T-beam" structure described by *Case (1922)* for the posterior cervical and dorsal vertebrae of *Desmatosuchus spurensis* and described as present in *S. robertsoni* (*Walker, 1961*) and *Typothorax* (*Martz, 2002*) and also occurs in *Paratypothorax* (*Martz et al., 2013* Fig. 9D). The right prezygapophysis is

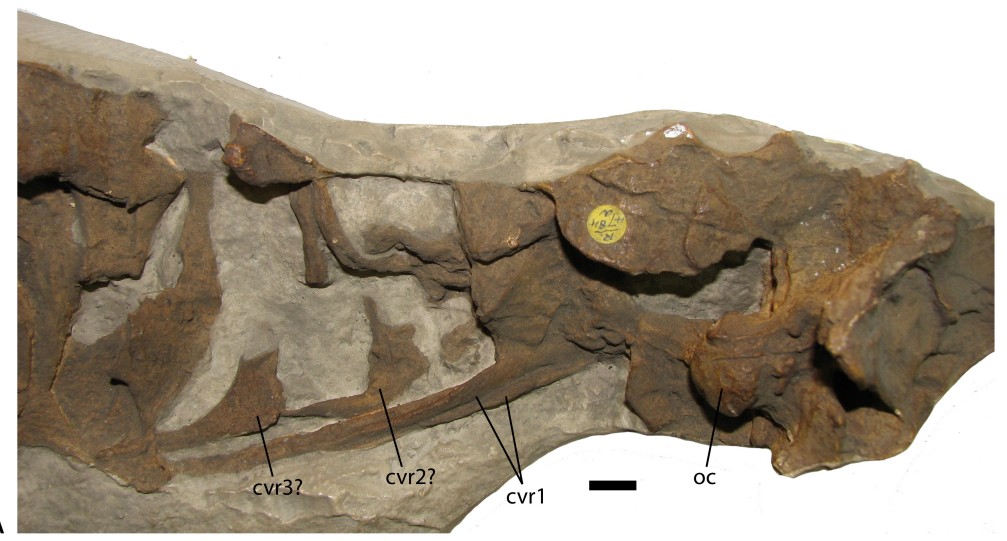

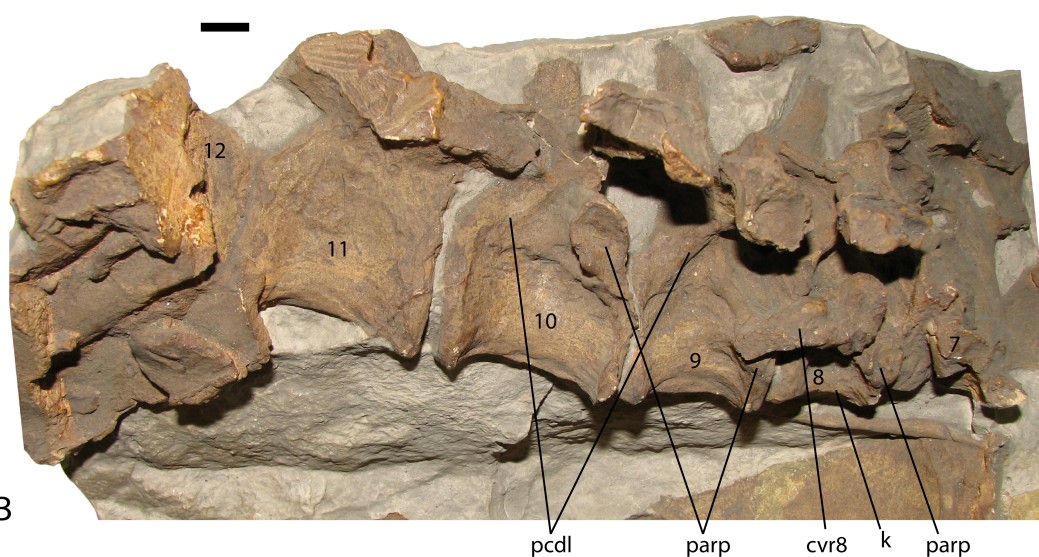

**Figure 4  Presacral vertebrae of *Stagonolepis robertsoni*** NHMUK PV R 4784a, casts of articulated sections of the presacral vertebral column in lateral view including the rear of the skull through the fourth cervical position (A), and the seventh through the twelfth presacral vertebrae (B). Scale bars equal 1 cm. Abbreviations: cvr, cervical rib; k, ventral keel; oc, occipital condyle of the basicranium; parp, parapophysis; pcdl, posterior centrodiapophyseal lamina.

present but not enough is present to tell if a hyposphene was present. The neural spine is present, but missing the apex.

The next three vertebrae are preserved in articulation (Fig. 4B). The parapophysis on the 9th cervical is still situated on the centrum, but slightly higher than its position in the preceding vertebra. The centrum is more elongate as well, measuring 31 mm.
Pre- and postzygadiapophyseal laminae (*Wilson, 1999*) are visible on the 9th and 10th presacrals (Fig. 4B). The 10th presacral is transitional between the cervical and dorsal series as previously described for *Desmatosuchus spurensis* (*Case, 1922*; *Parker, 2008a*; *Parker, 2008b*). The centrum is more elongate than the previous vertebra with a length of 36 mm. However, there is still a slight ventral keel as in the other cervicals, and although the parapophysis has migrated upwards onto the base of the neural arch, it is not located on the arch itself as in the trunk vertebrae. The neurocentral suture appears to be open.

The 11th presacral is the first true trunk vertebra as the parapophysis would now be situated on the posteroventral surface of the transverse process. Unfortunately this cannot be seen clearly as it is broken away (*Walker, 1961*: fig. 7i). The centrum has a length of 39 mm and is unkeeled. The 12th presacral is present, but covered by broken ribs. The cervical and trunk vertebrae of *S. robertsoni* lack oval depressions on the lateral sides just below the neurocentral sutures as in *Aetosauroides scagliai* (*Desojo & Ezcurra, 2011*).

## Caudal vertebrae

NHMUK PV R 4799b is a PVC cast of an isolated anterior caudal vertebra providing more details of the neural arch and spine (*Walker, 1961*: figs. 10c-e). The broad transverse processes (left equals 79 mm) do not extend ventral to the base of the centrum; the postzygapophyses are oriented at 45 degrees above horizontal. The centrum is blocky, with equant width and height of about 40 mm, but the entire vertebral height is 112 mm, with the neural spine contributing 40 mm to this measurement. Spinopostzygapophyseal laminae (*Wilson, 1999*) are present, as is the expanded neural spine table.

## Osteoderms

Numerous osteoderms of *Stagonolepis robertsoni* are preserved as natural molds; however, often they only produce partial casts. *Walker (1961)* discussed the osteoderms in what is now considered superficial terms, therefore one each of the best preserved dorsal paramedian and lateral osteoderms is redescribed determining estimated position using the technique presented by *Parker & Martz (2010)*. The dorsal paramedian osteoderm (NHMUK PV R 4790a; Fig. 5A) is from the left side, has a width/length ratio of 2.4/1, and an anastomosing surface patterning (*Taborda, Heckert & Desojo, 2015*), an anastomosing, interlaced network of high ridges surrounding circular and elongate pits closer to the posterior plate margin, and elongate, but irregular grooves on the anterior portion of the osteoderm. The width/length ratio, the slight ventral flexion, and the more medially situated position of the dorsal eminence suggests that this osteoderm is from the anterior caudal region based on comparison with the holotype specimen of *Calyptosuchus wellesi* (Case, 1932) and a referred specimen of *Aetosauroides scagliai* (MCP 13a-b-PV). The ornamentation is similar to the pattern in *Aetosaurus ferratus*, *Aetosauroides scagliai*, *Neoaetosauroides engaeus*, *Stagonolepis olenkae*, and *Paratypothorax andressorum* in that the ornamentation is non-radial posteromedially, but elongate and radial anteriorly along the full length of the anterior bar. However, in *S. robertsoni*, the ornamentation on the posterolateral region of the paramedians is distinctly non-radial and faint to absent along the entire posterior margin, whereas in the other taxa the paramedians possess very elongate

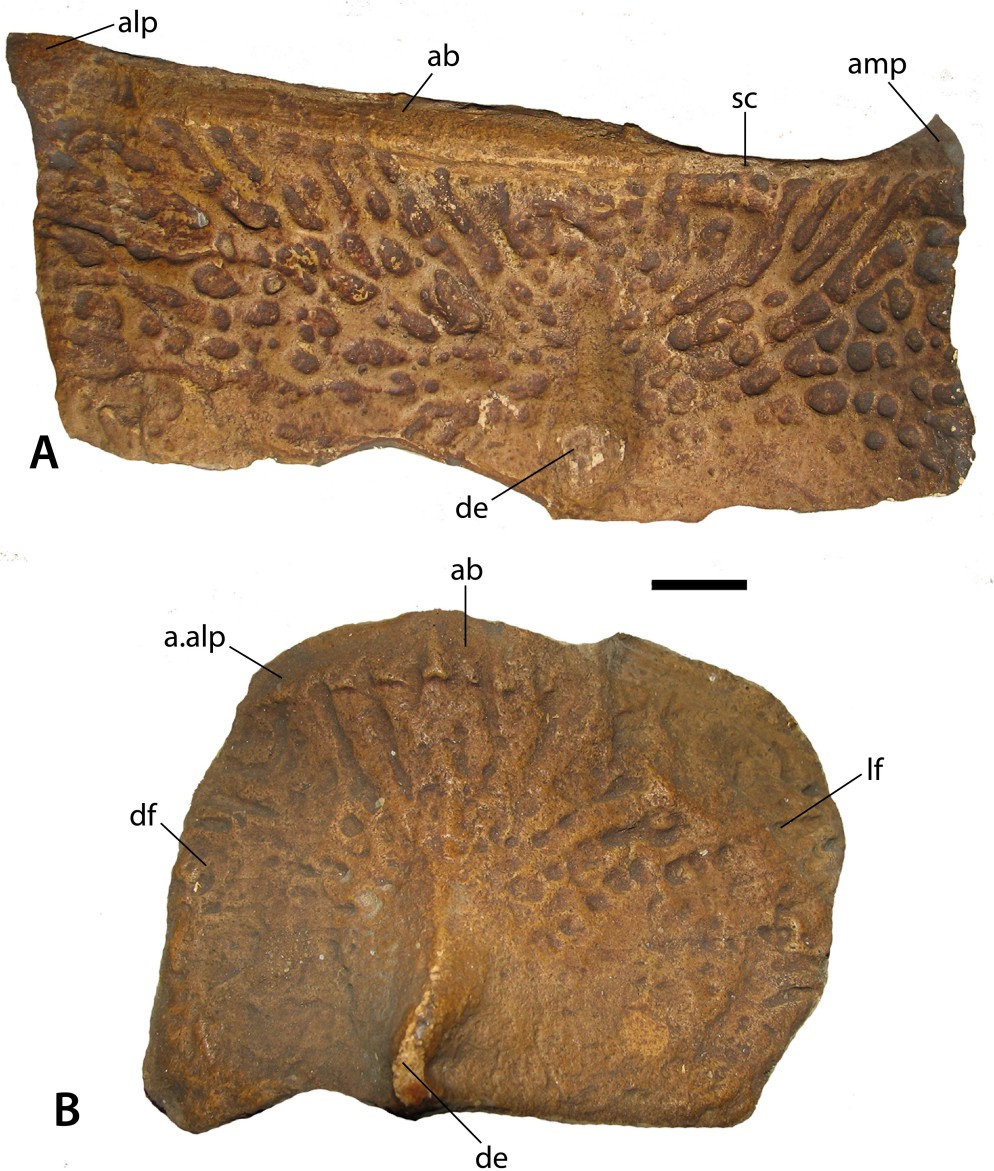

**Figure 5** **Casts of osteoderms of *Stagonolepis robertsoni*.** (A) left dorsal trunk paramedian (NHMUK PV R 4790a) in dorsal view; (B) right dorsal trunk lateral (NHMUK PV R 4789a) in dorsolateral view. Scale bar equals 1 cm. Abbreviations: a., articulation with listed element; ab, anterior bar; alp, anterolateral projection; amp, anteromedial projection; de, dorsal eminence; df, dorsal flange; lf, lateral flange; sc, scalloping of anterior bar margin.

grooves in the posterolateral portion of the paramedian osteoderm that are nearly parallel to the posterior margin.

This anastomosing ornamentation in *S. robertsoni* radiates from an elongate, but narrow, raised dorsal eminence that contacts the posterior osteoderm margin as in most non-desmatosuchine taxa that possess an eminence; in desmatosuchines, the eminence often contacts the posterior margin (*Parker, 2018*; *Parker & Martz, 2010*). This eminence is

rounded rather than distinctly pyramidal as in *Typothorax* (*Martz, 2002*) and *Longosuchus* (*Parker & Martz, 2010*), and offset medially from the center of the osteoderm as in nearly all aetosaurian posterior trunk and anterior caudal paramedians. The anterior portion of the osteoderm bears a raised, transverse, smooth strip of bone called the anterior bar (*Long & Ballew, 1985*) that bears anterolateral and anteromedial projections as in most non-desmatosuchine aetosaurs. The anterior bar maintains an even width across the lateral portion of the osteoderm, but thins significantly medially before expanding again at the anteromedial projection. This distinct medial thinning is termed 'scalloping' (following *Parker, 2016b*). This feature occurs in several other aetosaurians including *Aetosauroides scagliai* (PVL 2073), *Calyptosuchus wellesi* (UCMP 126844; *Parker, 2018*), *Scutarx deltatylus* (*Parker, 2016b*), and *Paratypothorax andressorum* (SMNS 5721). The medial edge is straight and the lateral edge slightly sinuous in dorsal view. In posterior view the osteoderm is moderately flexed.

The lateral trunk osteoderm (NHMUK PV R 4789a; Fig. 5B) is from the right side based on the presence of a distinct beveling of the anteromedial corner of the anterior bar, which represents an articulation surface for the anterolateral process of the adjacent paramedian osteoderm. The bar is thin but stretches across the entire anterior margin of the osteoderm. The osteoderm is trapezoidal in dorsolateral view and a ridge-like dorsal eminence that contacts the posterior margin divides the osteoderm into distinct dorsal and lateral flanges. The dorsal flange is roughly trapezoidal in dorsal view, whereas the lateral flange is slightly larger and sub-rectangular in dorsolateral view. The surface ornamentation is anastomosing and very faint in the posterior portion of the osteoderm, as seen in the paramedian. The osteoderm is slightly flexed ventrally and the angle between the two flanges is obtuse. In dorsolateral view the lateral margin is gently rounded. The medial margin is angled posteromedially, corresponding with the shape of the adjacent paramedian osteoderm.

The ventral osteoderms (NHMUK PV R 27404 [negative cast of holotype ELGNM 27R]; NHMUK PV R 4789a; Figs. 6A, 6B) are rectangular, bear an anterior bar, and possess an ornamentation of randomly arranged oblong pits, the 'drops' which *Agassiz (1844)* used to formulate the genus name. These differ from what is seen in *Calyptosuchus wellesi* (UCMP 27225; *Parker, 2018*), *Aetosauroides scagliai* (MCP13-a-b-PV; *Desojo & Ezcurra, 2011*), and *Typothorax coccinarum* (*Martz, 2002*; *Heckert et al., 2010*). These taxa also have rectangular ventral osteoderms, but the ornamentation arrangement is more radial and consists of very elongate furrows and ridges, even in *Typothorax*, where paramedian osteoderm ornamentation is distinctly pitted. *Coahomasuchus kahleorum* (*Heckert & Lucas, 1999*) and *Aetosaurus ferratus* (*Schoch, 2007*) preserve more equant, overlapping ventral osteoderms; but the surface ornament on these specimens is poorly preserved and difficult to comprehend. Where they are preserved, however, they appear to consist of finer pits, less densely packed, but nonetheless have a strong radial distribution as in *Coahomasuchus chathamensis* (*Heckert, Fraser & Schneider, 2017*) and the previous taxa. The ventral osteoderms of *S. robertsoni* also differ from *Stenomyti huangae* where the osteoderms are subrounded rather than square and broadly separated rather than overlapping (*Small & Martz, 2013*).

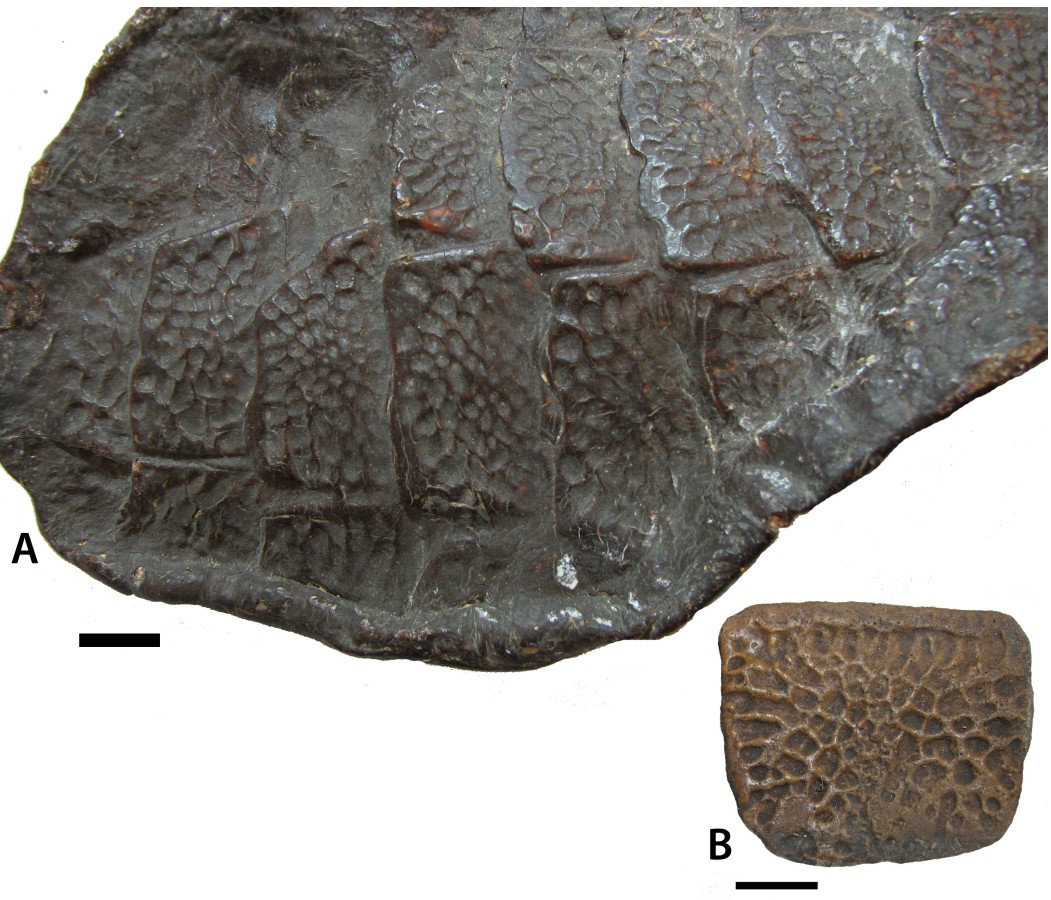

**Figure 6** **Ventral osteoderms of *Stagonolepis robertsoni*.** (A) close-up of NHMUK PV R 27404 (positive cast of EM 27R, the holotype specimen of *S. robertsoni Agassiz, 1844*) in ventrolateral view showing the detail of the overlapping osteoderms; (B) NHMUK PV R 4787a, referred ventral osteoderm in ventral view. Scale bar = 1 cm.

## Ilium

Described in detail by *Walker (1961)*, the ilium of *Stagonolepis robertsoni* is autapomorphic. The shapes and sizes of the processes of the iliac blade are particularly diagnostic for aetosaurian taxa. *S. robertsoni* possesses an elongate postacetabular blade that ends in an acute angled end in medial view (Fig. 7). Most other aetosaurians have a squared-off end of the postacetabular blade such as *Aetosauroides scagliai* (PVSJ 326; PVL 2073), *Calyptosuchus wellesi* (UCMP 32422), *Longosuchus meadei* (TMM 31185-40) and *Neoaetosauroides engaeus* (PVL 3525). *Typothorax coccinarum* (UMCP 35255) has a postacetabular blade that has a squared off end and is very short, barely extending past the posterior edge of the ischiadic peduncle.

The preacetabular blade of *S. robertsoni* differs from almost all aetosaurians in that it is very anteroposteriorly short in that it does not extend to the anterior margin of the pubic peduncle, very narrow, and ventrally hooked (Fig. 7). Most aetosaurians have preacetabular blades that extend anteriorly to the edge of the pubic peduncle, and are thick and triangular

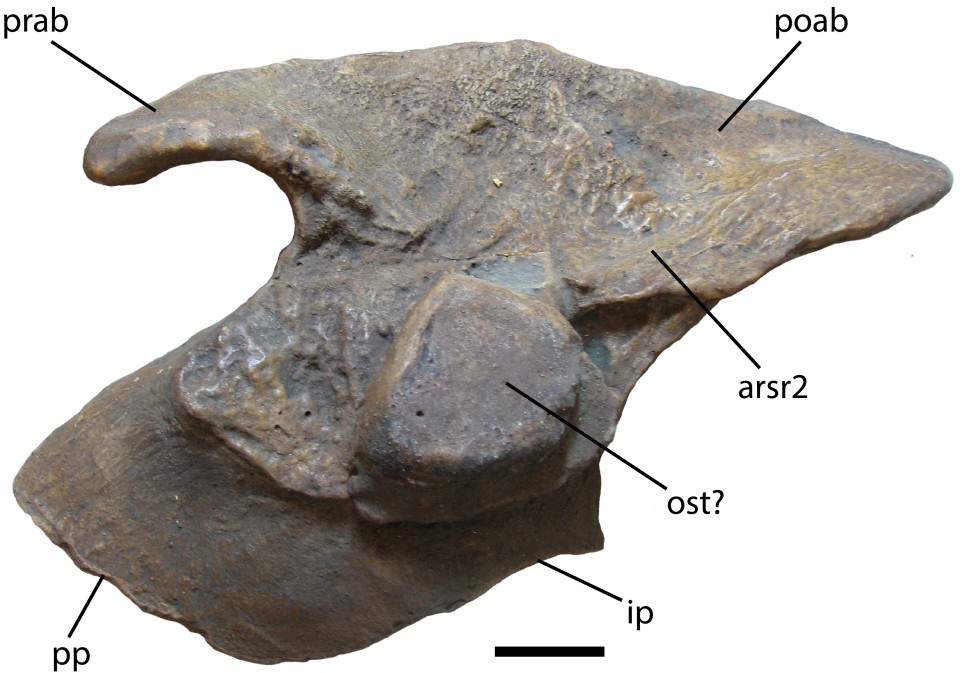

**Figure 7  Ilium of *Stagonolepis robertsoni*.** NHMUK PV R 4789. Cast of medial side of right ilium in medial view. Scale bar = 2 cm. Abbreviations: arsr2, articulation for sacral rib 2; ip, ischiadic peduncle; ost, osteoderm; poab, postacetabular blade; pp, pubic peduncle; prab, preacetabular blade.

in lateral view, such as *Aetosauroides scagliai* (PVL 2073), *Desmatosuchus spurensis* (UMMP 7476), and *Calyptosuchus wellesi* (UCMP 32422). *Neoaetosauroides engaeus* (PVL 3525) and *Typothorax coccinarum* (UCMP 35255) both have thin preacetabular blades and the element in *T. coccinarum* is very strongly hooked ventrally; however, in both taxa the preacetabular blades are much more elongate than in *S. robertsoni*, even extending past the anterior margin of the pubic peduncle.

Two ilia similar to that of *S. robertsoni* are TMM-31100-1 and UMMP 7322. TMM-31100-1, assigned to *Lucasuchus hunti* (*Long & Murry, 1995*), has the elongate and acute angled tip of the postacetabular blade, as well as a narrower and slightly ventrally hooked preacetabular blade; however, the preacetabular blade is more anteroposteriorly elongate than that of *S. robertsoni* extending to the margin of the pubic peduncle. UMMP 7322, referred to *Desmatosuchus spurensis* by *Long & Murry (1995)*, also has an acutely angled postacetabular blade end and a relatively narrow and slightly ventrally hooked preacetabular blade, but again the preacetabular blade is more elongate, in this case extending slightly anterior to the anterior margin of the public peduncle.

Overall the ilium most resembles that of *Aetosaurus ferratus* (*Schoch, 2007*) with a short, narrow and ventrally hooked preacetabular blade, but the posterior acetabular blade end is not as acutely angled in medial view.

### Pubis

An odd characteristic of the pubis of *S. robertsoni* is the presence of two large pubic foramina, the uppermost of which represents the obturator foramen (*Walker, 1961*). This was unique until the discovery of the same feature in the pubis of *Scutarx deltatylus* (*Parker, 2016b*). The pubis is poorly known in many aetosaurs; however, *Desmatosuchus spurensis* (MNA V9300) has only a single opening.

## PHYLOGENY

The most inclusive recent phylogenetic analysis of the Aetosauria is that of *Parker (2016a)*. That study is the only analysis to include both species of *Stagonolepis* (*S. robertsoni*; *S. olenkae*), and tests the relationships of other species that have historically been considered to belong to *Stagonolepis* (*Calyptosuchus wellesi*; *Aetosauroides scagliai*) (*Heckert & Lucas, 1999*; *Heckert & Lucas, 2000*; *Heckert & Lucas, 2002*). That analysis analyzed relationships between 26 in-group taxa utilizing 83 characters (*Parker, 2016a*). *Stagonolepis robertsoni* is recovered in a clade (Stagonolepidinae with *Polesinesuchus aurelioi Sawin, 1947*) within Desmatosuchia and as the sister taxon to Desmatosuchinae (Fig. 8). *Stagonolepis olenkae* and *Calyptosuchus wellesi* are recovered within Desmatosuchinae, whereas *Aetosauroides scagliai* is recovered outside of Stagonolepididae, as in other recent studies (*Desojo, Ezcurra & Kischlat, 2012*; *Heckert et al., 2015*).

*Stagonolepis sensu Sulej (2010)* is found to be paraphyletic, with *S. robertsoni* and *Polesinosuchus* forming a sister clade to *S. olenkae* and all other desmatosuchines. The previous published differences between *S. robertsoni* and *S. olenkae* all appear to be in the cranium (*Sulej, 2010*; *Antczak, 2016*; *Parker, 2016a*); however, many sections of the published description of the skull of *S. olenkae* (*Sulej, 2010*) are nearly verbatim to those published by *Walker (1961)* for *S. robertsoni*, so it is difficult to determine what material is actually being described in the Sulej paper.

*Antczak (2016)* used newly referred material from the Krasiejów quarry to hypothesize that that the two taxa may be conspecific, with some of the cranial differences between *S. robertsoni* and *S. olenkae* representing individual variation. Further support for this hypothesis was cited by *Antczak (2016)* as coming from the postcranial analysis of *S. olenkae* by *Lucas, Spielmann & Hunt (2007)* who also argued that *S. olenkae* was a synonym of *S. robertsoni*. However, this synonymy is based on plesiomorphies and not a detailed determination of apomorphies in the material. Moreover, the findings of *Antczak (2016)* suggest that the skull roof (ZPAL AbIII/466/17) proposed by *Sulej (2010)* may not serve adequately as a holotype for *S. olenkae*, making comparisons between the two taxa problematic.

Comparison of the paramedian osteoderms of the two taxa demonstrate that although both have similar width/length ratios, and anterior bars, and radial patterning, the patterning is quite distinct with that of *S. olenkae* consisting of more closely packed elongate ridges and grooves (Fig. 9). Nonetheless, this comparison is based on a single published osteoderm of *S. olenkae* (*Lucas, Spielmann & Hunt, 2007*:fig. 4a) and more osteoderm material is needed to further support any proposed differences. Thus, a full

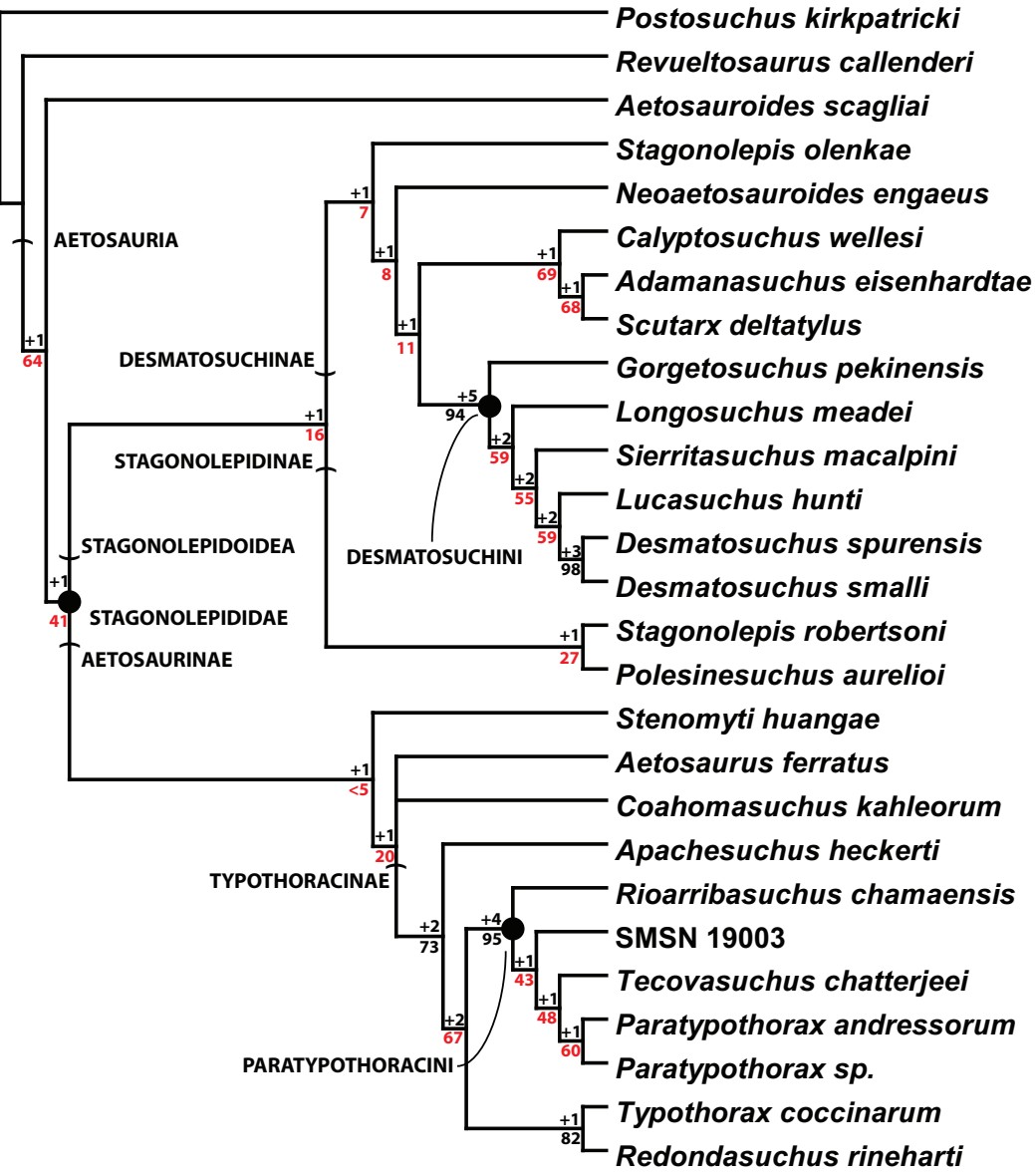

**Figure 8  Phylogenetic Analysis of the Aetosauria.** The reduced strict consensus tree from *Parker (2016a)*, with all named clades. Decay indices and bootstrap values are shown for all nodes, with bootstrap values under 70% (the confidence threshold of *Hillis & Bull, 1993*) shown in red.

description of the postcrania and osteoderms of *S. olenkae* is required to further examine potential differences between these two species.

Another recent phylogenetic analysis (Fig. 10; *Hoffman, Heckert & Zanno, 2018*) of the Aetosauria that also builds on successive analyses from *Heckert & Lucas (1999)*, *Parker (2007)*, *Desojo, Ezcurra & Kischlat (2012)*, *Heckert et al. (2015)*, and *Schoch & Desojo (2016)*, but does not include all currently known taxa (e.g., *Stagonolepis olenkae*) also recovers *S. robertsoni* in Stagonolepidinae as the sister taxon to Desmatosuchinae

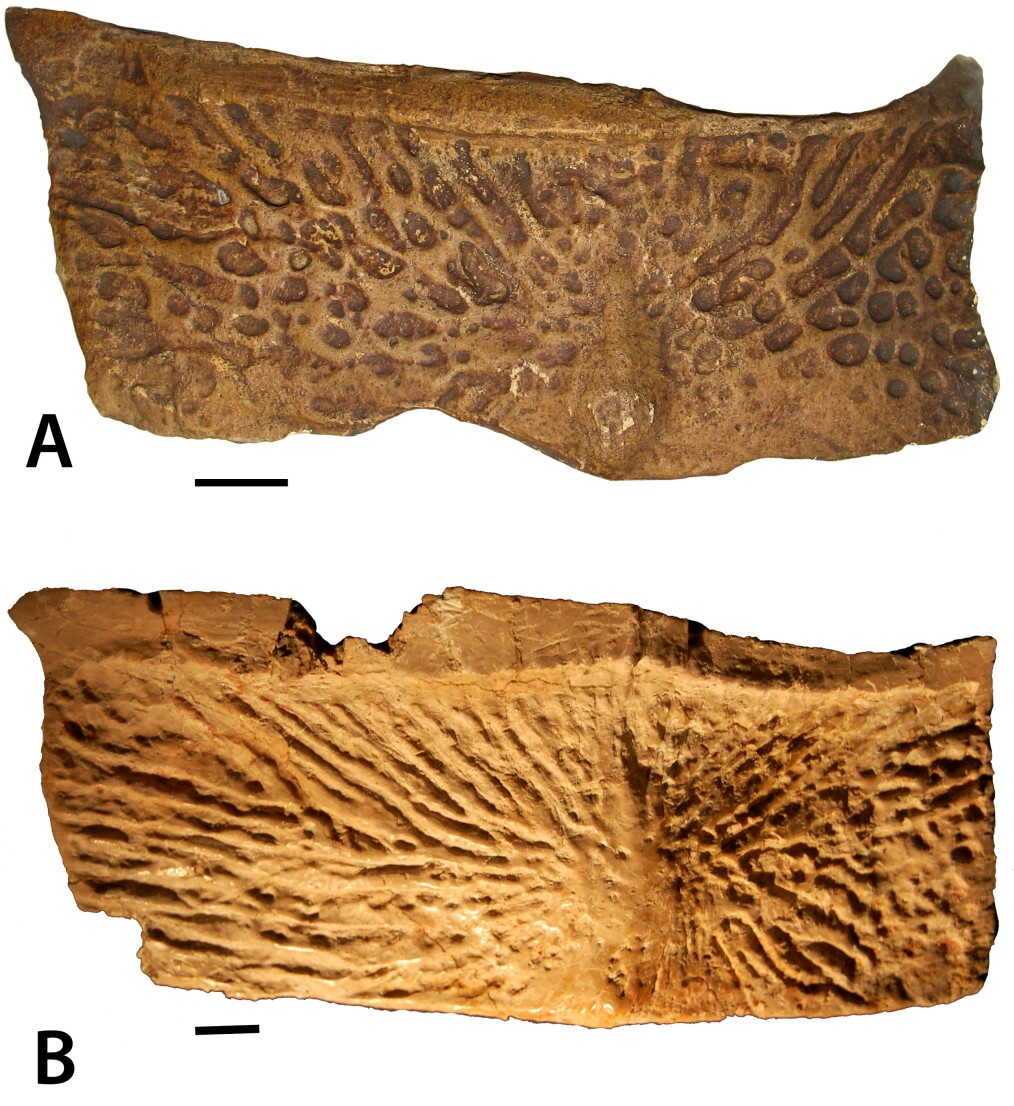

**Figure 9  Dorsal trunk paramedian osteoderms of *Stagonolepis*.** (A) *Stagonolepis robertsoni* NHMUK PV R 4787a, left dorsal trunk paramedian in dorsal view; (B) *Stagonolepis olenkae* PAN ZPAL AbIII 57011, left dorsal trunk paramedian in dorsal view. Scale bars = 1 cm.

(=Desmatosuchini of *Parker, 2016a*). Interestingly, with the exclusion of *S. olenkae* from the analysis, the sister taxon to *S. robertsoni* is *Calyptosuchus* (*Stagonolepis*) *wellesi*. Thus presently there is strong agreement between the phylogenetic position of *S. robertsoni* in these studies as more closely related to Desmatosuchinae than to Typothoracinae (Fig. 7, 9; *Parker, 2016a*; *Hoffman, Heckert & Zanno, 2018*).

## DISCUSSION

### Use of symplesiomorphies in aetosaurian taxonomy

The first formal diagnosis of *S. robertsoni* (*Heckert & Lucas, 2000*:1556) was based on plesiomorphies that do not even comprise a unique combination of characters within

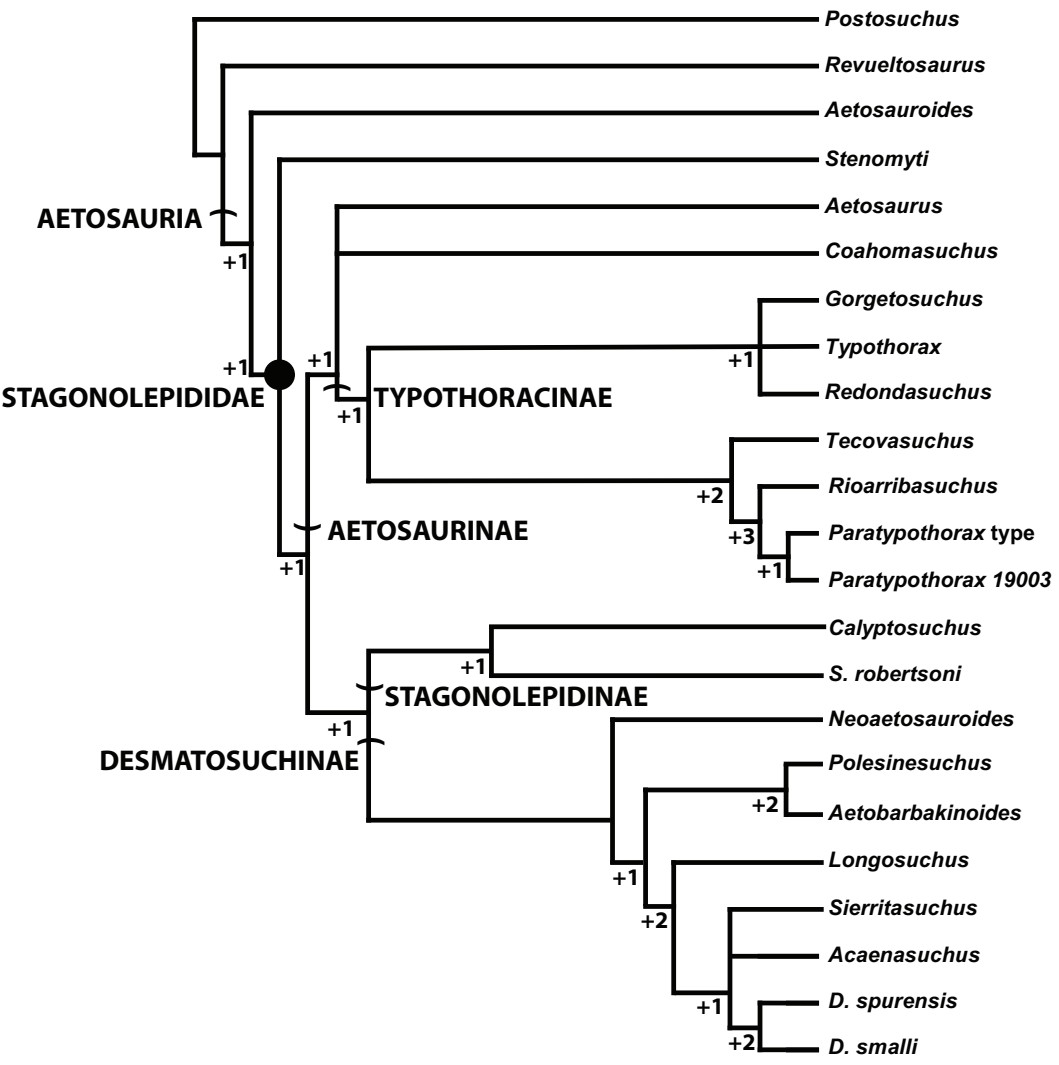

**Figure 10** Alternate phylogenetic hypothesis of the Aetosauria of *Hoffman, Heckert & Zanno (2018)* with clade names and decay indices for each node.

Aetosauria and was intended to only diagnose the genus. This conservative assessment of the taxon has allowed other specimens (e.g., *Aetosauroides scagliai*, *Calyptosuchus wellesi*) from other areas (North and South America) to be easily assigned to the genus, mainly for the purpose of building a global terrestrial vertebrate biostratigraphy for the Upper Triassic (e.g., *Lucas & Heckert, 1996*; *Heckert & Lucas, 1999*; *Heckert & Lucas, 2000*; *Heckert & Lucas, 2002*; *Lucas, Spielmann & Hunt, 2007*; *Lucas, 2017*). However, detailed comparison demonstrates that these taxa all bear unique combinations of characters that allow them to be differentiated (e.g., *Desojo & Ezcurra, 2011*; *Parker, 2018*). Assignment of specimens to genera can be subjective and based upon the taxonomic philosophy of the researcher. I have argued elsewhere (e.g., *Parker, 2018*) that utilizing monotypic genera in aetosaurian work can aid in removing some of the ambiguity regarding use of these taxa in

broader scale studies where genera are often used as a proxy for species. Diagnoses should be apomorphy-based, at the species level, or provide unique combinations of characters. Thus it is important that holotype specimens be reexamined utilizing discrete apomorphies in a phylogenetic context (*Nesbitt, Irmis & Parker, 2007*; *Irmis et al., 2007*; *Nesbitt & Stocker, 2008*; *Parker, 2013*).

This is especially true when taxa are being utilized for intercontinental biostratigraphic correlations (*Irmis et al., 2010*). For an example of how taxa may be assigned based on plesiomorphies for biostratigraphic correlation, a recent discussion on global biostratigraphy for the Late Triassic proposes a new hypothesis that *Neoaetosauroides engaeus* is a junior subjective synonym of *Aetosaurus ferratus* (*Lucas, 2017*). Support is given based on the width/length ratios and ornamentation patterns of the paramedian osteoderms (both symplesiomorphies for Aetosauria based on their presence in the non-stagonolepidid *Aetosauroides scagliai*) as well as an interpretation of similar sutural patterns of the skull based on published figures (*Lucas, 2017*:369). However, this interpretation ignores several key characters such as the presence of a laterally expanded premaxillary tip in *Neoaetosauroides engaeus* (*Desojo & Báez, 2007*), which is apomorphic for many aetosaurians (*Parker, 2016a*; *Hoffman, Heckert & Zanno, 2018*) although absent in *Aetosaurus ferratus* (*Schoch, 2007*) as well as *Aetosauroides scagliai*, which polarizes the character distribution. Among other character differences with *A. ferratus*, *N. engaeus* also lacks a well-developed antorbital fossa. Thus, synonomy between the two taxa is based on overall general similarity and symplesiomorphy rather than discrete synapomorphies, and currently unsupported. The practice of using plesiomorphic characters to assign specimens and to synonymize taxa has long been out of favor in vertebrate paleontology and should no longer be acceptable (e.g., *Sereno, 1990*; *Padian, Lindberg & Polly, 1994*; see further discussion in *Nesbitt & Stocker, 2008*). Data that utilize this approach should not used for broader scale studies until the synapomorphy distributions among included taxa are fully evaluated and supported.

## Status of the holotype of *S. robertsoni*

The holotype of *S. robertsoni* is the impression of a fragment of the ventral carapace of a single specimen (ELGNM 27R) that shows several partial rows and columns of imbricated predominantly rectangular osteoderms (Fig. 11; *Huxley, 1877*:pl.1, fig. 1). These osteoderms have an anterior bar and a surface pattern of numerous drop-shaped pits radiating from the osteoderm center; hence the name *Stagonolepis*, which means "drop scale". Since the initial descriptions by *Huxley (1877)* and *Walker (1961)* subsequent authors have assigned other species to *Stagonolepis robertsoni* based on similarities of the dorsal paramedian osteoderms (e.g., *Murry & Long, 1989*; *Long & Murry, 1995*; *Lucas & Heckert, 2001*; *Heckert & Lucas, 2002*; *Lucas et al., 2007b*). However, none of these authors have addressed the diagnostic status of the type specimen.

Ventral osteoderms are known from several other aetosaurian taxa including *Coahomasuchus kahleorum*, *Calyptosuchus wellesi*, *Scutarx deltatylus*, *Neoaetosauroides engaeus*, *Aetosaurus ferratus*, and even the non-aetosaurian *Revueltosaurus callenderi*. However, as described above presently the ornamentation in the ventral osteoderms of the

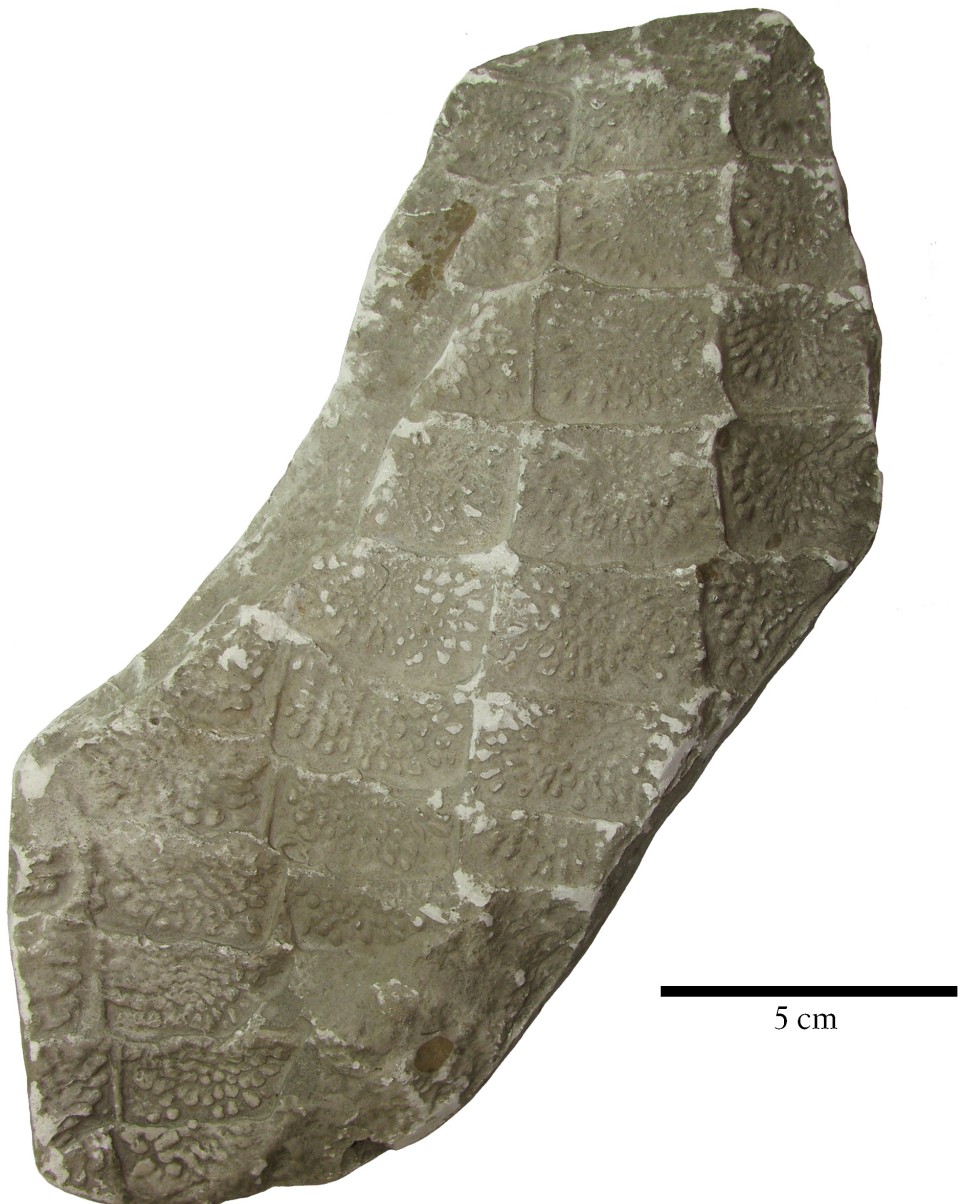

**Figure 11  Holotype specimen of *Stagonolepis robertsoni*.** Cast of EM 27R, the holotype specimen of *Stagonolepis robertsoni Agassiz, 1844*, a series of imbricated osteoderms from the ventral trunk region.

holotype specimen (ELGNM 27R) of *S. robertsoni* differs from that of other aetosaurians in that it consists of randomly arranged oblong pits separated by a latticework of thin ridges. These pits are tightly packed and cover the majority of the osteoderm surface. This differs from what is seen in other aetosaurian taxa where the ornamentation consists of narrow grooves and oblong pits radiating from a central point on the osteoderm, and presently *S. robertsoni* is currently valid based on this character as well as the unique combination of characters listed and discussed above. A similar patterning is found in what is thought to be

ventral osteoderms in the purported aetosaurian *Chilenosuchus forttae Casamiquela, 1980* (*Desojo, 2003*); however, the aetosaurian affinities of this taxon are uncertain (*Desojo et al., 2013*). Regardless even if *C. forttae* is shown to unambiguously represent an aetosaur, it can still be distinguished from *S. robertsoni* by the ornamentation of the dorsal osteoderms, which lack a radial pattern in *C. forttae* (*Desojo, 2003*). However, this might necessitate the designation of a neotype specimen to conserve the name *Stagonolepis robertsoni* as the holotype specimen will no longer be diagnostic and the taxon valid only based on a unique combination of characters (*Parker, 2008b*; *Parker, 2014*).

## CONCLUSIONS

*S. robertsoni* is the oldest named aetosaurian and as such has long served as the standard for aetosaurian osteology. Despite this status, the material is difficult to work with because of its preservation and because the only sites where *S. robertsoni* fossils have been recovered are no longer active. However, the holotype specimen, although fragmentary, is presently diagnostic and *Walker*'s (*1961*) description is extremely faithful to the existing material and still serves as the basis for our understanding of this taxon.

In this contribution I redescribe and note several features that have recently become important characters for the purpose of comparing aetosaurians taxonomically and phylogenetically. *S. robertsoni* has a paramedian osteoderm morphology that bears a unique combination of characters including a raised anterior bar with anteromedial, anterior, and anterolateral processes (projections), a 'scalloped' anterior edge of the anterior bar medial to the anterior process, a short anterolateral process, a posteriorly placed, pyramidal dorsal eminence, and an anastomosing pattern of pits and grooves radiating from the eminence that lacks elongate nearly parallel grooves. *S. robertsoni* is also the only aetosaurian to preserve an extremely elongate first cervical rib, which possibly is an autapomorphy of the taxon. Posterior cervical vertebrae are keeled ventrally and bear diapophyseal and zygapophyseal laminae. The parabasisphenoid is anteroposteriorly elongate with significant separation between the basitubera and the basipterygoid processes. The ventral osteoderms have an autapomorphic ornamentation.

These characters serve to differentiate *S. robertsoni* from all other aetosaurs including those who are or have been historically assigned to the same genus including *Stagonolepis olenkae*, *Calyptosuchus wellesi*, and *Aetosauroides scagliai*. These assignments were made based on general similarity and symplesiomorphies rather than synapomorphies, a practice that is widely discouraged in vertebrate paleontology.

**Institutional Abbreviations**

| | |
|---|---|
| ELGNM | Elgin Museum, Elgin, Scotland |
| MCP | Museo de Ciencias e Tecnología, Porto Alegre, Brazil |
| MACN | Museo Museo Argentino de Ciencias Naturales 'Bernardino Rivadavia', Buenos Aires, Argentina |
| MCZD | University of Aberdeen Zoology Department, Aberdeen, Scotland |
| MNA | Museum of Northern Arizona, Flagstaff, Arizona, USA |
| NHMUK | Natural History Museum, United Kingdom |

| | |
|---|---|
| **PEFO** | Petrified Forest National Park, Petrified Forest, Arizona, USA |
| **PVL** | Paleontología de Vertebrados, Instituto ''Miguel Lillo,'' San Miguel de Tucumán, Argentina |
| **PVSJ** | División de Paleontologia de Vertebrados del Museo de Ciencias Naturales y Universidad Nacional de San Juan, San Juan, Argentina |
| **TMM** | Texas Vertebrate Paleontology Collections, University of Texas, Austin, Texas, USA |
| **UCMP** | University of California Museum of Paleontology, Berkeley, CA, USA |
| **UMMP** | University of Michigan Museum of Paleontology, Ann Arbor, Michigan, USA |
| **USFM** | Universidade Federal de Santa Maria, Santa Maria, Brazil |
| **YPM** | Yale Peabody Museum of Natural History, New Haven, Connecticut, USA |
| **ZPAL** | Institute of Paleobiology, Polish Academy of Sciences, Warsaw, Poland |

## ACKNOWLEDGEMENTS

Thank you to Sandra Chapman (NHMUK), Lorna Steele (NHMUK), David Gower (NHMUK), Julia Desojo (MACN), Mark Goodwin (UCMP), Kevin Padian (UCMP), the late Jaime Powell (PVL); Ricardo Martinez (PVSJ), David & Janet Gillette (MNA), and Matthew Smith (PEFO) for access to specimens and data under their control as well as discussions. Thank you to Jeffrey Martz for discussions. An earlier version of the manuscript was completed under the partial requirements of a dissertation at the University of Texas by the senior author. Careful reviews by Richard Butler, Michael Benton, Julia Desojo, Jeffrey Martz, and academic editor Mark Young improved the manuscript. This is Petrified Forest National Park Paleontological Contribution #56. The work presented here is that of the author and does not represent the views or opinions of the United States Government.

### Funding

Financial assistance for this project was provided by the Jackson School of Geosciences at the University of Texas at Austin, the Lundelius Fund, the Francis L. Whitney Endowed Presidential Scholarship, the Ronald K. DeFord Scholarship Fund, and the Systematics Association. The funders had no role in study design, data collection and analysis, decision to publish, or preparation of the manuscript.

### Grant Disclosures

The following grant information was disclosed by the author:
Jackson School of Geosciences at the University of Texas at Austin.
Lundelius Fund.
Francis L. Whitney Endowed Presidential Scholarship.
Ronald K. DeFord Scholarship Fund.
Systematics Association.

## Competing Interests

The author declares there are no competing interests.

## Author Contributions

- William G. Parker conceived and designed the experiments, performed the experiments, analyzed the data, contributed reagents/materials/analysis tools, prepared figures and/or tables, authored or reviewed drafts of the paper, approved the final draft.

## Data Availability

The specimens described in this manuscript are at the Natural History Museum of London (accession numbers appear in the Materials section).

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
