# Peer review of "Anatomical notes and discussion of the first described aetosaur Stagonolepis robertsoni (Archosauria: Suchia) from the Upper Triassic of Europe, and the use of plesiomorphies in aetosaur biochronology"

_PeerJ, doi:10.7717/peerj.5455_

## Round 0.1 · original submission · Minor Revisions

Dear authors,

I am sorry for the delay in this decision. I have accepted the decision of 'minor revision' from the three reviewers.

I have some additional comments that the authors should address prior to resubmission (in addition to those made by the reviewers):
1. Authority and date should be provided for each species-level taxon at first mention. Please ensure that the nominal authority is also included in the reference list.
2. Specimens from the NHMUK. The NHMUK uses PV (palaeo vertebrates in their specimen numbers for fossil vertebrates), and either 'R' or 'OR' followed by a space in front of the number itself (for fossil reptiles). For example, I suspect NHMUK 4789a should read as: NHMUK PV OR 4789a; and NHMUK R4787 would be NHMUK PV R 4587. However, it would be best if you contact the relevant curator to double-check.
3. Please replace 'tree' with 'cladogram'.

Please note that PeerJ does not do a full linguistic check (such as for the French issues noted by reviewer two). That is the responsibility of the authors.

Once again, thank you for submitting your manuscript to PeerJ and I look forward to receiving your revised submission.

·

Basic reporting

The manuscript by Parker & Martz is a solid description and discussion of some previously overlooked anatomical features of Stagonolepis, a classic aetosaur taxon from the Late Triassic of Scotland, with a discussion of the taxonomy and phylogenetic position of this taxon. There are no substantial problems with this work - it is generally well written and adequately referenced and appropriately organised. I have a number of minor comments on typographical errors and grammar that are marked directly on the attached PDF.

Experimental design

This paper contains original research, and there are no problems with research design.

Validity of the findings

Results and conclusions are well supported by the presented data.

Additional comments

I don't have any substantial edits to suggest. I have made a number of minor comments on typographical errors and grammar on the attached PDF. There are also some minor comments on terminology and the correct names and abbreviations of repositories for the described fossil specimens.

·

Basic reporting

The paper provides an update on the earlier classic monograph by Walker (1961), and confirms that that older work was largely accurate. It might then seem unnecessary to publish a further paper when there is no new material. However, since 1961, many new aetosaurian taxa have been found and, in particular, these have been subject to several cladistics analyses – the current authors have been world leaders in this endeavour, and so their comments on the classic material are enlightening. The restudy has allowed the authors to confirm the correct coding of phylogenetically informative characters in Stagonolepis robertsoni, and provide a diagnosis. They further resolve issues including mis-assignment of some American material to Stagonolepis, a mis-assignment that had been used as a biostratigraphic indicator, but which now fails. Detailed comparisons are given between S. robertsoni and other aetosaurs, most of which were not known in 1961.

In particular, the osteoderms are described more fully, in terms not considered important in 1961, but now critical in determining the phylogenetic position of the taxon.

Experimental design

Anatomical description, identification of phylogenetically informative characters, and phylogenetic analysis all properly carried out.

Validity of the findings

Findings are all well justified.

Additional comments

37 the despite = despite
77 is the = is to
81 instead = but instead
90 are in = is in
106 NHMUK R4787; Sandstone = NHMUK R4787, sandstone
122–128 insert taxonomic commas, after author names
308 36mm = 36 mm
407 S. robertsoni = S. robertsoni
421 uppermost which = uppermost of which
435 Stagonolepididae as = Stagonolepididae, as
485 apomorphy based = apomorphy-based
513 drop shaped = drop-shaped
533 status = status,
587 poisons = poisons
587 grés = grès
587 Dévonian = Dévonien
587 ed de = et de [if you don’t know French, it’s best simply to copy and paste from a reputable source: https://scholar.google.co.uk/scholar?hl=en&as_sdt=0%2C5&q=Monographie+des+poisons+fossiles+du+vieux+grés+rouge+ou+Système+Dévonian+%28Old+Red+Sandstone%29+des+Isles+Britanniques+ed+de+Russie.&btnG=.]
753 Triassic Reptiles from the Elgin Area = Triassic reptiles from the Elgin area

·

Basic reporting

The MS by Parker and Martz is a competent analysis of the traditional key aetosaur taxon Stagolepis robertsoni that deserves to be published in the “Peerj” after several modifications suggested below.

It is also important to stress that I am not an English native speaker, so I did not preformed an in depth review on the use of the English language, which seems mostly adequate. Yet, I would be me more comfortable if an English native speaker (the other reviewer or the editor) could perform such a revision.

Experimental design

no comment

Validity of the findings

no comment

Additional comments

In the introduction should be mention the new material from the USA assigned to Stagonolepis based on the osteoderm morphology (e.g. patron and type of ornamentation, proportions L/W in Long & Murry 1985, Lucas&Heckert 2001) and the new species S. olenkae described by Sulej 2010 to the same genus from Poland.

In the Material section should be mention a table with all the information from the specimen analyzed in this contribution were referred to Stagonolepis robersoni. (e.g. NHMUK R4787 a cast- left side of the skull and mandibule)

The diagnosis should be check , because some autopomorphies mention by the authors are share with other aetosaur taxa, such as:

There are some autopomorphy characters from the diagnosis and combination that also distinguished to be check:

-The randomly arranged (reticular patron) of ornamentation described in Stagonolepis robertoni by the authors is also present in Chilenosuchus forttae (Casamiquela 1980, Desojo 2003). Although the validity of Chilenosuchus as an aetosaur is debate (see Desojo et al. 2013), it should be include in the comparisons and discussion section of the present manuscript.

-The autapomorphy: trunk paramedian osteoderms lack ornament along the posterior edge in the region of the dorsal eminence should be described in more detail and check in more aetosaur taxa, because it vary among the differ regions of the dorsal armor of one specimen (e.g. cervical, dorsal, sacral, caudal regions). see Taborda et al. 2015, Schoch & Desojo 2016 Fig.2 A-D.

- Character from the osteoderm onramentation should be compared with Chilenosuchus and Aetosauroides scagliai (see attach file).

The osteoderm description section (line 328) should be improve with more comparison with Chilenosuchus (see Desojo 2003) for the ornamentation pattron and ventral osteoderm comparison, as well as with Aetosauroides scagliai (see Taborda et al. 2015) for the anastomosing ornamentation definition (previously used by Taborda et al) and the dorsal eminence variation among paramedian osteoderms of a single carapace. This variation of ornamentation of the paramedian dorsal eminence affects the diagnosis provided by the authors here.

line 492: This statement is very interesting, but no forget that the ontogenetic variation could be work here,
Minor comments:
The first mention in the text of Stagonolepis olenkae is between quotation marks, could you justify it?

line 227: no really, it is possible to observe in S. olenkae and Aetosauroides scagliai new material (UFSM 11050) on the left side

line 230: also present in S. olenkae

line 248: it is difficult to see in the figure 3. Could you indicate it? This dorsal protuberance is also present in other aetosaurs (e.g. Paratypothorax, Neoaetosauroides), please check it.

line 271: also in Aetosauorides scagliai (see Taborda et al. 2015 PVL 2059)

line 287: a well development ventral keel is present in Neoaetosauorides (Desojo&Baez 2005) and absent in Aetobarbakinoides (Desojo et.al. 2012). so, I am not sure about your statement (291) of its variability. More information about intraspecific variation should be done to confirm or not it.

line 295: this lamina was described by Desojo & Ezcurra 2011 (pag. 599) as present in S. robersoni and A. scagliai , among other aetosaurs, named posterior infradiapophyseal lamina; please check.
There are several references in the text but not in the references section (e.g. Reese 1915, Heckert et al. 2010, etc.), please, check. Also, few citations are no in the text but in the references section, and the correct year of publication for Benton & Walker is 2011.

Fig. 1A: There is a bone on the right side of the right surangular indet, could be a right angular? please, check.
Could you indicate the 5th premaxillary tooth crown that you mentioned?

Fig.2a: Could you indicate de foramina? at least 4 foramina are present on the maxilla body.
Should be good if you can indicate the six alveoli in Fig.2B
2B: could you indicate the interdental plates and the alveolus?

Fig 3: Could you indicate the specimen number and the Premaxilla/maxilla contact? Also, should be indicate the premaxillary 4 teeth.

---

## Round 0.2 · Minor Revisions

Dear author,

I have decided to give a decision of 'minor revisions' rather than 'accept', as I would like to give you a chance to respond to the two comments of reviewer 3.

I look forward to receiving your revised version.

·

Basic reporting

Nothing more to add.

Experimental design

Nothing more to add.

Validity of the findings

Nothing more to add.

Additional comments

None.

·

Basic reporting

no comment

Experimental design

no comment

Validity of the findings

no comment

Additional comments

The authors did a great job with the suggestions and modifications. Congratulations.
Only two comments:

1-my original comment: line 227: no really, it is possible to observe in S. olenkae and Aetosauroides scagliai new material (UFSM 11050) on the left side
-authors replied: This material has not been formally published and is being worked on by someone else so I have not included it.

I understand and respect the position, Thank you. On the other hand, actually the manuscript is in press:
Osteology of the first skull of Aetosauroides scagliai Casamiquela 1960 (Archosauria: Aetosauria) from the Upper Triassic of southern Brazil (Hyperodapedon Assemblage Zone) and its phylogenetic importance
PONE-D-17-29053R2
Dear Ana Carolina Biacchi Brust,
We are pleased to inform you that your manuscript has been judged scientifically suitable for publication and will be formally accepted for publication once it complies with all outstanding technical requirements.
Within one week, you will receive an e-mail containing information on the amendments required prior to publication. When all required modifications have been addressed, you will receive a formal acceptance letter and your manuscript will proceed to our production department and be scheduled for publication.
Shortly after the formal acceptance letter is sent, an invoice for payment will follow. To ensure an efficient production and billing process, please log into Editorial Manager at https://www.editorialmanager.com/pone/, click the "Update My Information" link at the top of the page, and update your user information. If you have any billing related questions, please contact our Author Billing department directly at authorbilling@plos.org.
If your institution or institutions have a press office, please notify them about your upcoming paper to enable them to help maximize its impact. If they will be preparing press materials for this manuscript, you must inform our press team as soon as possible and no later than 48 hours after receiving the formal acceptance. Your manuscript will remain under strict press embargo until 2 pm Eastern Time on the date of publication. For more information, please contact onepress@plos.org.
With kind regards,
Thierry Smith, Ph.D.
Academic Editor
PLOS ONE


2-In the introduction should be mention the new material from the USA assigned to Stagonolepis based on the osteoderm morphology (e.g. patron and type of ornamentation, proportions L/W in Long & Murry 1985, Lucas&Heckert 2001) and the new species S. olenkae described by Sulej 2010 to the same genus from Poland.
-authors replied: This has been dealt with in many previously papers and I did not want to burden the reader with a complete rehash of the taxonomic history of all of the genera previously synonymized with S. robertsoni since it had already been covered several times in other recent papers. Instead I just dealt with the history of the Scottish specimens of interest.

-It is really a pity, because the topic of this paper is specifically on the taxonomy and anatomy of Stagonolepis robertsoni, so, a short taxonomy account of the genus and species should be improve it.

---

## Round 0.3 · accepted · Accept

Dear author,

Many thanks for your revised manuscript. After reading it, I have accepted it for publication in PeerJ.

Once again, thank you for submitting your manuscript to PeerJ and I hope you will use us again as your publication venue.

If we need to clarify any details required to move the manuscript forward, then our production staff will get in touch with you. Otherwise, a proof will be forthcoming shortly for your review.

Congratulations and thank you for your submission.

#